# The Binding of the SARS-CoV-2 Spike Protein to Platelet Factor 4: A Proposed Mechanism for the Generation of Pathogenic Antibodies

**DOI:** 10.3390/biom14030245

**Published:** 2024-02-20

**Authors:** Thi-Huong Nguyen, Li-Yu Chen, Nida Zaman Khan, Annerose Lindenbauer, Van-Chien Bui, Peter F. Zipfel, Doris Heinrich

**Affiliations:** 1Institute for Bioprocessing and Analytical Measurement Techniques (iba), 37308 Heilbad Heiligenstadt, Germany; 2Faculty of Mathematics and Natural Sciences, Technische Universität Ilmenau, 98694 Ilmenau, Germany; 3Institute of Miccrobiology, Friedrich-Schiller-University, 07745 Jena, Germany; 4Department of Water Supply and Wastewater Treatment, Eichsfeldwerke GmbH, 37308 Heilbad Heiligenstadt, Germany; 5Fraunhofer Institut für Silicatforschung, Neunerplatz, 97082 Würzburg, Germany

**Keywords:** SARS-CoV-2 spike protein, PF4, PF4/SP antibodies, generation, mechanism

## Abstract

Pathogenic platelet factor 4 (PF4) antibodies contributed to the abnormal coagulation profiles in COVID-19 and vaccinated patients. However, the mechanism of what triggers the body to produce these antibodies has not yet been clarified. Similar patterns and many comparable features between the COVID-19 virus and heparin-induced thrombocytopenia (HIT) have been reported. Previously, we identified a new mechanism of autoimmunity in HIT in which PF4-antibodies self-clustered PF4 and exposed binding epitopes for other pathogenic PF4/eparin antibodies. Here, we first proved that the SARS-CoV-2 spike protein (SP) also binds to PF4. The binding was evidenced by the increase in mass and optical intensity as observed through quartz crystal microbalance and immunosorbent assay, while the switching of the surface zeta potential caused by protein interactions and binding affinity of PF4-SP were evaluated by dynamic light scattering and isothermal spectral shift analysis. Based on our results, we proposed a mechanism for the generation of PF4 antibodies in COVID-19 patients. We further validated the changes in zeta potential and interaction affinity between PF4 and SP and found that their binding mechanism differs from ACE2–SP binding. Importantly, the PF4/SP complexes facilitate the binding of anti-PF4/Heparin antibodies. Our findings offer a fresh perspective on PF4 engagement with the SARS-CoV-2 SP, illuminating the role of PF4/SP complexes in severe thrombotic events.

## 1. Introduction

Since the start of the global pandemic in December 2019, the newly appeared novel Severe–Acute Respiratory Syndrome-Corona-virus-2 (SARS-CoV-2 or COVID-19) has infected approx. 772 million cases and caused approx. 6.96 million deaths globally [1]. Hospitalized patients with COVID-19 are at high risk for developing thrombotic complications such as venous thromboembolism (VTE), stroke, and limb ischemia [2]. This increased the risk of in-hospital mortality [3], as around 71% of the deaths had abnormal coagulation profiles [3,4]. The incidence of thromboembolism in patients with severe coronavirus pneumonia is 25% [5,6]. Severe COVID-19 is characterized by a prothrombotic state associated with thrombocytopenia, with microvascular thrombosis being almost invariably present in the lungs and other organs at postmortem examination [7].

At the beginning of the COVID-19 pandemic, Warkentin et al. reported some similar patterns and many comparable features between the COVID-19 virus and heparin-induced thrombocytopenia (HIT) [8]. For example, both HIT and COVID-19 show similar characteristics such as moderate/mild thrombocytopenia, a reduction in platelet count and d-Dimer, a thrombotic rate of ~50%, clots in venous more than in arterial, and Limb ischemic syndrome is about 5% or 1%, respectively [8].

HIT develops in up to 3% of patients with heparin (H) administration after major surgery, even when the low molecular weight heparin is used [9,10]. Both HIT and COVID-19 patients including COVID-19 patients in an intensive care unit (ICU) cause approximately 50% thrombosis frequency [8,11]. Later, several studies detected anti-platelet factor 4 (PF4)/H antibodies in these patients using the widely used PF4/H enzyme-linked immunosorbent assay (ELISA) [12,13]. However, among these patients, only a very few sera activate platelets, indicating a mixture of PF4/H antibodies of different reactivities, were detected in these cases. It became clearer when a recent study detected anti-PF4 antibodies including IgG, IgM, and IgA in 95% of hospitalized COVID-19 patients, irrespective of prior heparin treatment [14]. These antibodies significantly reduced platelet counts in the circulating system during hospitalization, indicating their clinical relevance. The level of anti-PF4 antibodies was found to increase with increasing the disease severity score. The pathogenic antibodies contain Fc residues that activate platelets via platelet FcγRIIa receptors and are associated with disease severity and pulmonary pathology [13,15].

Severe COVID-19-involved patients suffer from extremely high levels of thrombotic complications. Heparins are recommended for the management of coagulopathy [16], even at higher/double doses [17], as they appear to be associated with better prognosis in severe COVID-19 patients. This means that COVID-19 patients with heparin therapy also suffer from HIT in which platelet-activating anti-PF4/H antibodies are generated. Furthermore, vaccine-induced thrombotic thrombocytopenia (VITT) has been confirmed due to the development of platelet-activating PF4-antibodies or HIT-like antibodies after COVID-19 vaccination, causing severe thrombotic thrombocytopenia [18,19,20,21]. More seriously, very sick COVID-19 patients exhibited a combination of high levels of super-active antibodies and super-activated neutrophils, which are destructive and can explode white blood cells [22]. Although HIT-like antibodies have been detected, the mechanism of the development of these antibodies in both COVID-19 and VITT patients has not yet been elucidated. It is unclear until now what triggers the body to produce a high frequency of these antibodies in COVID-19-involved patients [12].

In contrast to VITT and COVID-19, the mechanism of HIT has been clearly understood [23,24,25,26]. A delineation of the mechanisms of COVID-19 disease remains a high priority, as it may foster the development of increasingly effective therapeutic strategies. HIT develops when the immune system responds to the ultra-large immunocomplexes that are formed between the positively charged chemokine PF4 (CXCL4) and the negatively charged heparin [27,28,29]. Immune cells, especially B lymphocytes, produce PF4/H antibodies (HIT Abs) against the formed PF4/H complexes. HIT antibodies contain Fc fragments that can cross-link and activate platelets, monocytes, neutrophils [30], and bind to endothelial cells [31,32,33]. Bound PF4 to heparan sulfate on endothelial cells leads to tissue factor expression that activates monocytes while activated platelets produce platelet-derived microparticles that accelerate thrombin generation, resulting in an increased risk for thrombosis. This can induce the most frequent immune-mediated adverse drug reactions, the life-threatening autoimmune HIT [34]. Recently, we identified a new mechanism of autoimmunity mediated by anti-PF4 antibodies (aPF4 Abs) [35]. These antibodies have a high binding affinity to PF4 as they cluster PF4 molecules and form PF4/antibody complexes that expose binding epitopes for typical PF4/H antibodies. Based on the knowledge gained in the HIT system, we hypothesize that SARS-CoV-2 spike protein (SP) can similarly cluster PF4 molecules as aPF4 Abs do, forming PF4/SP complexes to promote immune cells producing PF4/SP antibodies or HIT-like antibodies. In this study, we proved the binding of SARS-CoV-2 SP to PF4 by utilizing quartz crystal microbalance (QCM), dynamic light scattering (DLS), isothermal spectral shift analysis (ISSA), and enzyme-linked immunosorbent assay (ELISA). Our results allowed us to propose a mechanism for the development of PF4/SP antibodies caused by the response of immune cells to the formed PF4/SP complexes in COVID-19 and VITT patients.

## 2. Methods

### 2.1. Ethics

The use of human sera was approved by the ethics board of Thüringen.

### 2.2. Reagents

The following reagents were used: full-length SARS-CoV-2 spike protein of original COVID-19, purity > 90% (Biozol, Eching, Germany); lyophilized human PF4 isolated from human platelets (Chromatec, Greifswald, Germany); QCM sensors (Biolin Scientific, Darmstadt, Germany); ACE2 (Biorbyt Ltd., Cambridge, UK); capillary zeta cell (Malvern Instruments Ltd., Malvern, UK); human IgG and KKO (Thermofisher, Karlsruhe, Germany); anti-mouse IgG HRP (Biozol, Eching, Germany).

### 2.3. Quartz Crystal Microbalance (QCM)

All preparations and experiments were conducted at room temperature (RT). The quartz sensor QSX 301 with a resonance frequency of 4.95 MHz ± 50 kHz (Biolin Scientific Darmstadt, Germany) was cleaned in a 5:1:1 mix of H_2_O:NH_3_:H_2_O_2_ solution in an ultrasonic bath for 10 min. After rinsing with water and ethanol and drying with nitrogen, a self-assembled monolayer (SAM) of cysteamine and glutaraldehyde which contains functional aldehyde (-CHO) groups for the binding of protein was formed on the sensors. Spike protein of 2.5 µg/mL (or PF4 of 20 µg/mL) was covalently immobilized on the SAM layer for 15 min before blocking free aldehyde (-CHO) groups with 1M ethanolamine for 1 h. After rinsing with PBS, PF4, ACE2, or human IgG, samples up to 10 µg/mL were added for binding at a pumping speed of 500 µL/min with an incubation time of 10 min. The real-time resonant frequency change was recorded at the third overtone due to stability constraints at a higher order. QCM real-time resonant frequency changes were observed on Qsoft software (version 2.5.22.707, Q sense, Biolin Scientific, Europe) and analyzed using the Sauerbrey equation through Qtools software (version 3, Quantum Design, Darmstadt, Germany) and Origin 2023.

### 2.4. Dynamic Light Scattering (DLS)

To test if SP clusters PF4, SP up to 7.7 µg/mL was titrated in a folded capillary zeta cell (Malvern Instruments Ltd., Malvern, UK) containing PF4 of 20 µg/mL in water (pH 7.4, conductivity of 0.318 mS/cm) using our previous protocol testing with HIT antibodies [35]. For controls, PF4 was replaced by human IgG isolated from healthy donors (Thermofisher, Germany) as a negative control while ACE2 (Biorbyt Ltd., Cambridge, UK) was used as a positive control. Measurements were carried out in water at 20 °C and light scattering was detected at 173° using the Zetasizer Nano-S system (Malvern Instruments Ltd., Malvern, UK). The migration speed in an electric field was assessed with DLS for 10 repetitions. Data analysis was performed using the Zetasizer software, version 7.11 (Malvern Instruments Ltd., Malvern, UK), and Origin 2023.

### 2.5. Isothermal Spectral Shift Analysis (ISSA)

The equilibrium dissociation constant (K_D_) of the interaction between SP and PF4 was measured using the Monolith-X (Nanotemper Technologies, München, Germany) as described previously [36]. Covalent labeling of lysine residues of PF4s was performed using the Protein Labeling Kit RED-NHS 2nd Generation (cat# MO-L011; NanoTemper Technologies GmbH, Germany). Briefly, 100 μL of a 10 μM PF4 or ACE2 solution was mixed with a threefold molar excess of dye and incubated for 30 min at room temperature in the dark. Unreacted dye was removed using the B-Column of the labeling kit, and the labeled proteins were eluted from the column using PBS. Labeled PF4 or ACE2 concentrations were measured with a spectrophotometer (DS-11, DeNovix, Wilmington, DE 19810, USA). For K_D_ measurements, PF4 concentration was kept at 100 nM while the unlabeled SP was serially diluted (16 dilutions) from 1.725 µM to 26 pM in PBS without calcium and magnesium containing 0.01% Tween20. For ACE2–SP binding, ACE2 concentrations from 75 to 225 nM were mixed with SP from 2.75 μM to 83 pM. The samples were loaded into Premium Capillaries (cat# MO-K025, NanoTemper Technologies, Germany), and the spectral shift measurement was performed at 25 °C using the Monolith-X instrument (NanoTemper Technologies, Germany) at 100% excitation power and medium IR laser power. Ratios of the fluorescence intensities at 670 and 650 nm were used for the analysis of all ligand concentrations using the MO.Control software (v2.5.3, NanoTemper Technologies GmbH, Germany). 

### 2.6. Enzyme-Linked Immunosorbent Assay (ELISA)

ELISA was performed by coating SPs of 10 µg/mL in DPBS overnight at 4 °C on a 96-well plate or non-coated plates as controls (Table 1) and then blocked with 7.5% goat serum as previously described [37,38,39]. After rinsing five times with DPBS, 100 µL PF4 of 20 µg/mL or only buffer as control (Table 1) was incubated on SP-coated plates for 1 h at RT for binding. Afterward, unbound PF4 molecules were removed and anti-PF4 mouse antibody (KKO) (Thermofisher, Germany) of 1 µg/mL or only buffer as control (Table 1) was added for binding to PF4/SP. Subsequently, 100 anti-mouse antibody IgG HRP (Biozol, Germany), 1:50,000 dilution in DPBS, or only buffer as control (Table 1) was added and incubated for 1 h at RT. Wells were washed five times with DPBS and 100 TMB solution was added for 5 min before stopping the reaction using 100 μL H_2_SO_4_. 

With human HIT sera, three well-characterized HIT sera containing PF4/H antibodies were tested. These antibodies only bind to PF4/H complexes but not to PF4 alone coated on ELISA plates [40]. Here, PF4/SP (20 µg/mL PF4 and 10 µg/mL SP) instead of PF4/H complexes were prepared as described above. After rinsing, dilution of HIT sera (1:200) was added to the PF4/SP complexes-coated plate for binding. After rinsing with PBS, 100 μL anti-human IgG HRP (Biozol, Germany) (1:20,000 dilution) was incubated for 1 h at RT. Finally, 100 TMB solution was added for 5 min before stopping the reaction using 100 μL H_2_SO_4,_ and absorbance was recorded at 450 nm. Data analysis was performed using Excel or Origin 2023.

## 3. Results 

### 3.1. The Binding of SARS-CoV-2 SP to PF4 by the Mean of the Quartz Crystal Microbalance (QCM)

We first utilized quartz crystal microbalance (QCM), which is a label-free and highly sensitive technique, to detect the binding of SP with PF4 based on additional mass or frequency changes on a real-time basis. The SP was first immobilized on a QCM chip via glutaraldehyde linkers, and pumping PF4 on the chip allowed for their bindings (Figure 1A). As controls, human IgG isolated from healthy donors which are known to not interact with SPs was used as a negative control whereas the ACE2 that interacts strongly with SP served as a positive control. Typical QCM spectra showed a strong frequency shift when adding IgG control or PF4 to the SP-coated sensors (Figure 1B). The titration of proteins up to 10 µg/mL including PF4 to SAM layer or PF4, IgG, and ACE2 to SP-coated sensors allowed us to determine the variations in mass changes (Figure 1C). The strongest mass increase was observed for ACE2 (Figure 1C, blue), followed by PF4 (Figure 1C, red), and the lowest for human IgG (Figure 1C, violet) when they were added to the SP-coated sensors. Pumping PF4 to sensors coated with SAM shows only background mass changes (Figure 1C, black), which is comparable with that of IgG to the SP-coated chip (Figure 1C, violet). The higher mass increase of ACE2 than PF4 when binding to the same SP-coated sensors was due to its higher molecular weight (85 kDa for ACE2 vs. 32 kDa for PF4 tetramer). The results together indicate the binding of PF4 to SP.

### 3.2. The Binding of SARS-CoV-2 SP to PF4 Alters Protein Structures

To further understand the binding of SP to PF4, we examined the interaction in the liquid phase using dynamic light scattering (DLS) instead of immobilization on the solid phase. For that, SP (up to 7.7 µg/mL) was titrated in a cuvette containing PF4 (20 µg/mL), and the change in surface zeta potential was determined (Figure 2A). The SP or PF4 alone has a zeta potential around −16 mV or +12 mV, respectively. As only a surface potential around zero facilitates protein aggregation, this negative or positive zeta potential indicates their low tendency of accumulation among SP or PF4 (=high stability). When titrating SP concentrations into PF4, the zeta potential of PF4 was switched from positive to negative values. At low SP concentrations (≤1 µg/mL), zeta potential largely varied, indicating an unstable protein mixture. However, the system became stable as zeta potential saturated at higher SP concentration (from ≥2.5 µg/mL). Compared with SP alone, PF4/SP complexes exhibited higher in negative zeta potential (Δ_ZP_ ≈ −10 mV) (Figure 2B, arrow). The results indicate that SP bound to PF4 and altered structures of the original proteins, resulting in the exposure of residues with negatively charged surfaces. With controls, the SP did not cause a significant change in zeta potential when interacting with human IgG (Figure 2C), whereas it caused a strong increase in negative zeta potential when interacting with ACE2 (Figure 2D). However, the binding mechanisms of the SP to PF4 differs from that to ACE2. SP promoted PF4 to expose amino acids with negative zeta potential that reached saturation at approx. 1.0 µg/mL SP (about −25 mV) (Figure 2B). However, its binding caused an expression of amino acids with positive zeta potential, indicated by a reduction in negative zeta potential as SP concentration increases (Figure 2D). Interestingly, PF4–SP binding reached a saturation from 1.0 µg/mL but no such saturation was seen in the ACE2–SP system up to 7.7 µg/mL SP concentration (Figure 2D), indicating a higher binding affinity of SP to PF4 than ACE2. 

To further understand these interactions, we determined their equilibrium dissociation constant (K_D_) using isothermal spectral shift analysis (ISSA) [36]. The fluorescent signal in ISSA is recorded simultaneously only at two pre-selected wavelengths of 670 and 650 nm with photon-multiplier tubes. The ISSA consumes a lower amount of the sample (only a few microliters of the sample at low nanomolar concentrations per data point). PF4s were fluorescently labelled, whereas the labeling of the target SP molecules was not required. Label-free SP avoids any effect when it binds to PF4. In the titration of the SP of different concentrations into a PF4 of fixed concentration, ISSA measures the affinity of the interaction between SP and PF4 by detecting variations in the fluorescent signals at 670 and 650 nm. Ratios of the fluorescence intensities at 670 and 650 nm were used for the analysis. By fitting fluorescence signal ratios measured at various SP concentrations, the K_D_ of PF4–SP binding could be determined: K_D_ = 586 ± 185 nM (Figure 2E). Interestingly, the K_D_ of the SP–PF4 interaction was lower than that of SP–ACE2, K_D_ = 2080 ± 20 nM in our ISAA experiments, indicating a higher binding affinity of SP to PF4 than to ACE2. The results are consistent with the observation of changes in zeta potential obtained in DLS experiments (Figure 2B,D). 

### 3.3. The Confirmation of PF4–SP Binding in the Enzyme-Linked Immunosorbent Assay (ELISA)

To further confirm the binding of SP to PF4, we performed a sandwich ELISA. For this, SP (10 µg/mL) was immobilized on a 96-well ELISA to capture PF4 (10 µg/mL), while an ani-PF4 HIT-like antibody (KKO, 1 µg/mL) was linked to PF4. A secondary antibody conjugate with an enzyme as the detection molecule was linked to the KKO (Figure 3A). By adding TMB substrate, the chromogenic reaction converted the enzyme into a colored product which could be measured as the optical density (OD) at the absorbance of 450 nm using a plate reader. Different controls were carried out to confirm PF4/SP binding (Figure 3B). Goat serum was used to block nonspecific bindings in all samples. At the standard OD boundary, which is normally used to distinguish between positive and negative results in ELISA for HIT, all controls showed OD < 0.5 while SP bound to PF4 displayed OD > 0.5 (Figure 3B, orange arrow). However, the OD for the PF4 control appears relatively high, although it predominantly remains below the established cutoff of OD = 0.5. This observation arises from the inherent binding of KKO to PF4 alone. Notably, upon the formation of PF4/SP complexes, a significantly enhanced binding of KKO compared to its interaction with PF4 alone was observed. The results again emphasized the binding of SP to PF4. 

It is known that the PF4/H antibodies bind only to PF4/H complexes (not to PF4 alone), as the binding of heparin to PF4 in these complexes causes the formation of binding epitopes for antibodies [35]. This allows for the detection of HIT antibodies via PF4/H-coated substrates [41] or even in a solution [42]. Recently, HIT-like antibodies have been detected in VITT and COVID-19 patients using PF4/H ELISA, and therefore,we used PF4/H antibodies to understand if the SP also causes a change in PF4 conformation that facilitates the binding of HIT antibodies as heparin does. For that, we coated the SP first on 96-well plates. After blocking the free surfaces that may allow for the binding of PF4, PF4 was added to the wells for the formation of PF4/SP complexes. Well-characterized sera containing PF4/H antibodies which have already been confirmed to have OD > 0.5 were then added for bindings to PF4/SP complexes. Interestingly, the tested antibodies bound strongly to the formed PF4/SP complexes (OD >> 0.5) (Figure 3B, red arrow), whereas the human IgG control bound weakly (OD < 0.5) (Figure 3B, green arrow), indicating that PF4/SP complexes expose binding epitopes for HIT antibodies. 

## 4. Discussion

We proved that SP binds/clusters PF4 in vitro. Even though a previous study has quantified using ELISA that SP binds to PF4 [43], ELISA results only confirm PF4/SP binding but does not provide insights into its binding mechanism. Our study surpasses prior research by providing compelling evidence of PF4 changes within the PF4/SP complex, as revealed through dynamic light scattering. In particular, the SP induced conformational changes in PF4 that switched the surface zeta potential of PF4 from positive to negative values. Employing four methods, we not only validate the presence of the PF4/SP complex but also underscore the dynamics of the interaction between PF4 and SP. Crucially, the binding mechanism between PF4 and SP stands out from that of ACE2–SP, exhibiting distinct features in terms of conformational changes and interaction affinity. Significantly, we reveal a groundbreaking discovery in which the PF4/SP complexes facilitate the binding of anti-PF4/H antibodies, albeit not in precisely the same manner as observed with the PF4/H complex. This novel finding is absent in the current literature and adds a distinctive dimension to our understanding. These results contribute a fresh perspective to understanding how PF4 engages with the SARS-CoV-2 spike protein and shed light on the implications of PF4/SP complexes in amplifying thrombotic events. 

Recently, monoclonal antibodies (mAbs) targeting the SARS-CoV-2 spike protein have been established to block viral entry into cells. Monoclonal antibodies represent a vital therapeutic choice for individuals grappling with severe cases of COVID-19, offering a lifeline in particular to high-risk individuals for whom vaccination may not be a feasible or immediate option [44]. The mAbs are designed to neutralize the virus by binding to the spike protein on its surface. However, the binding of PF4 to the spike protein might block mAbs from accessing the same spike protein targets. This may lead to an unsuccessful treatment of infected patients using mAbs. Our results indicate that PF4–SP binding can be a reason to explain the evasion and resistance of viruses/mutants to mAbs reported recently [44,45,46]. 

In clinical aspects, some similar patterns and many comparable features between COVID-19 and HIT patients have been observed [8,11]. In HIT, the immune cells respond to the ultra-large PF4/H complexes that trigger the generation of anti-PF4/H antibodies (Figure 4A) [31]. Based on the known mechanism of HIT and our confirmation of binding between SP and PF4 as well as the presence of PF4- or HIT-like antibodies detected in VITT and COVID-19 patients [12,13,14,21], we propose a mechanism for the generation of these antibodies in patients (Figure 4B). In the HIT system, the size of ultra-large PF4/H immunocomplexes can be up to several micrometers [27,47,48]. As opposed to the SARS-CoV-2 with a size of 60–140 nm [49], the cross-linking among PF4 molecules and viruses can also develop ultra-large immune complexes. These PF4/virus particles can promote immune cells, like B-cells, to generate PF4-antibodies in a similar way that ultra-large antigenic PF4/H complexes do (Figure 4B). This explains how the body generates platelet-activating anti-PF4 antibodies in COVID-19 patients, as detected recently [12,13]. Furthermore, PF4 has a high affinity to polyanions on many cell surfaces [34] such as monocytes, neutrophils, endothelial [31], and even cancer cells [50,51]. Thus, PF4/virus particles have a high potential to mediate and even activate platelets and other cells while PF4-antibodies are known to activate human platelets, monocytes, and neutrophils [31]. Activated platelets release high concentrations of PF4 that further facilitate the generation of additional antibodies, enhancing severe thrombosis formation over time (Figure 4C). We conclude that the synergy of high PF4 concentration, PF4/virus complexes, and PF4-antibodies or the synergistic COVID-HIT system majorly contributed to the development of unusual severe thrombotic thrombocytopenia in COVID-19 patients (Figure 4C). However, the degree of complication may occur differently depending on the amount of PF4, and the quantity of the virus in each individual, and the unique biological backgrounds. 

Platelet-activating PF4-antibodies have been detected also in COVID-19-vaccinated individuals even without heparin exposure [19,21,52]. Vaccine-induced thrombotic thrombocytopenia (VITT) is attributed to the reactivity of PF4-antibodies. In the working process of vaccines, soluble and partial spike proteins could be generated. We speculate that soluble and partial spike proteins can enter the blood circulation, forming PF4/SP complexes to trigger the formation of PF4-antibodies. It has been reported that VITT patients’ sera increased platelet activation over time, counted from the first day of being vaccinated with the AstraZeneca vaccine [19]. An explanation for this is that the amount of soluble and partial spike proteins are increasingly produced over time, resulting in a stronger response of immune cells to PF4/SP that increases the production of PF4-antibodies. As multiple complex factors including PF4/virus and PF4/soluble-SP can develop in COVID-19 patients, a high concentration of PF4-antibodies can be developed. Consistently, when tested with VITT patients’ sera, a high OD ~3 was obtained in PF4/H ELISA [19] while HIT sera [37] shows OD only approx. 2.

Even though this study describes a novel observation of binding between SARS-CoV-2 SP and a PF4 that provides a significant impact on global health, it still meets several limitations. Firstly, Swank et al. [53] reported that the concentration of SP antigen in patients is only at the pg/mL level; however, our tested concentrations in this study were in the μg/mL range. Due to the detection limit of the used methodologies, we were unable to test the bindings at low SP concentrations. Under static conditions in our study, measurements with low SP concentrations led to low-detection signals which even fall into the background; thus, we must amplify the signal by increasing protein concentration. The situation differs from the condition in the blood circulation due to the presence of high PF4 concentration and shear force. It has been reported that healthy donor sera contain less than 1.0 µg/mL PF4, but PF4 concentration in the plasma of both severe and non-severe COVID-19 patients strongly increased even up to 30 µg/mL [54]. We hypothesize that high PF4 concentration under blood shear force promotes the interaction between PF4 and even low concentrations of SP. Furthermore, PF4 most likely binds not only to the soluble spike proteins that could be detected in patients (at pg/mL level) but also interacts with those directly on living viruses in the circulation. As a result, both PF4/soluble-SP and PF4/virus complexes may trigger the formation of PF4/SP antibodies. Therefore, unlike our tests under static conditions in vitro, the possibility that PF4 interacts with SP in vivo is enhanced under blood flow conditions. It is important for future studies to test the binding of PF4 to low SP concentrations’ underflow like in the blood circulation to realize whether PF4–SP bindings occur. For this, utilizing advanced technologies that provide higher resolution for the detection of weak signals initiated by only a few PF4–SP interactions would help to understand whether only low SP concentration in patients is sufficient to trigger the formation of antigenic PF4/SP complexes. Secondly, notable variability is observed in the error bars during the assessment of PF4/SP complex binding with sera from HIT patients (Figure 3). This analysis was conducted using only the sera from three well-characterized HIT patients, suggesting the necessity for a more extensive examination involving a larger cohort of HIT sera to unequivocally validate the interaction between HIT antibodies and PF4/SP complexes. Furthermore, it is imperative to undertake the additional optimization of the system to delineate the optimal conditions that ensure the highest stability of PF4/SP complexes. This comprehensive approach will enhance the reliability and significance of our findings. Thirdly, while this study concentrated on confirming the PF4/SP binding, a vital next step involves a thorough examination of the specific SP component interacting with PF4. Moreover, analysis of the interaction between PF4 and SP at molecular level using structural analytical techniques such as X-ray crystallography or nuclear magnetic resonance (NMR) would provide valuable insights into the arrangement, bond distances, and dynamics of the interaction. This investigation could yield insights for future drug design, enabling the development of targeted therapeutics with enhanced precision and efficacy. Recently, modulating SARS-CoV-2 reactivity through electric fields has been reported as a pathway to innovative therapies [55]. Identifying the precise region within SP that engages with PF4 holds the key to advancing pharmaceutical strategies for PF4-related disorders or even with the application of electric fields. Fourthly, in this study, we are unable to confirm whether COVID-19 and VITT patients’ sera contain antibodies against PF4/SP complexes. This is due to the fact that only a limited number of COVID-19 and VITT patients who developed pathogenic HIT-like antibodies could be confirmed. It is challenging to collect enough samples for our tests; especially, COVID-19 patients have not always been confirmed since the middle of 2022. It is important to collect a significant number of samples for systematic testing. Additionally, we tested the PF4 binding to only soluble spike proteins, not the whole virus. As the spike protein dynamically undergoes conformational changes while entering the cells via binding to ACE2 receptors, the question of whether or not the binding ability of PF4 to the conformationally changed SP is enhanced during the process of viral entry could not be addressed in our study. By understanding this process, we can clarify if PF4s affect the process of SARS-CoV-2 viral entry. Lastly, more details on conformational changes and experiments in vivo/ex vivo require further efforts. Further, an evaluation of the effect of different strains in the context of PF4 binding is essential for a comprehensive understanding of this field. In light of the rapid evolution of SARS-CoV-2 and the emergence of diverse variants, it is paramount to underscore the relevance of our findings within the evolving landscape of the pandemic. While our investigation focused on the interaction between PF4 and the original SP of COVID-19, it is importance to consider variant-specific interactions for a comprehensive understanding. The original SP, being a representative of the early stages of the pandemic, provides valuable insights into the initial dynamics of PF4 interactions. However, the virus has undergone genetic changes leading to the emergence of variants with distinct characteristics. To further enhance the applicability of our study, future research endeavors will benefit from systematically exploring interactions with various SARS-CoV-2 spike protein variants. This approach ensures that our findings remain pertinent and contribute meaningfully to the broader understanding of the complex interplay between the virus and host factors. By embracing the dynamic nature of the pandemic, our study aims to provide a foundation for ongoing investigations that can adapt to the evolving viral landscape, ultimately contributing to more effective strategies for diagnosis, treatment, and prevention. 

## Figures and Tables

**Figure 1 biomolecules-14-00245-f001:**
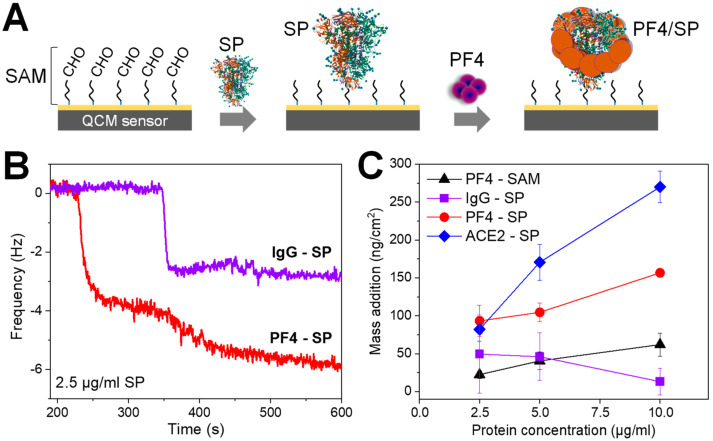
Interaction between PF4 and SP in QCM. (**A**) Illustrations show the immobilization of SP on a QCM chip via a self-assembled monolayer (SAM) layer composed of cysteamine (blue) and glutaraldehyde linkers (black). Pumping PF4 (or other proteins) to the chip allows for the binding of PF4 to SP coated on the sensor. (**B**) Typical QCM spectra show frequency shift differences when IgG control (**violet**) or PF4 (**red**) was added to the SP-coated sensor. (**C**) Titration of proteins up to 10 µg/mL shows the strongest increase in mass addition for ACE2 (**blue**), followed by PF4 (**red**), and the lowest for human IgG (**violet**) when interacting with SP-coated sensors; PF4 to SAM without SP (**black**) as another control shows only background binding which is comparable with IgG to SP-coated chip (**violet**). Data are shown as mean and SD (*n* = 2–3).

**Figure 2 biomolecules-14-00245-f002:**
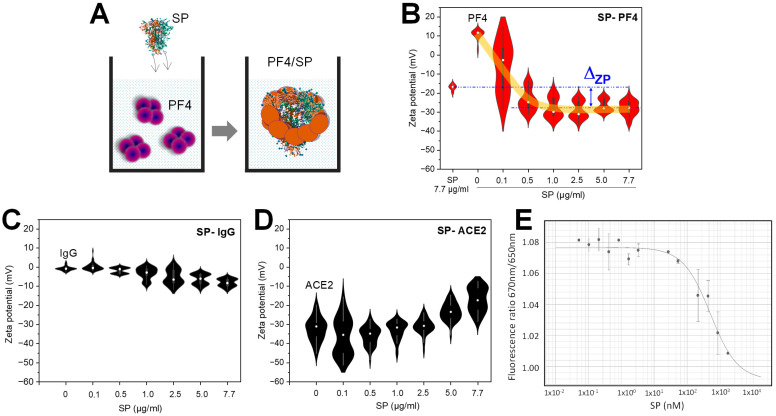
Binding of SP to PF4 alters protein structures determined by DLS ISSA. (**A**) Cartoons demonstrate titration of SP into a DLS cuvette containing PF4: binding of SP to PF4 caused changes in PF4 structure (indicated by color changes from violet to orange). (**B**) Titration of SP (zeta potential approx. −16 mV) into PF4 (zeta potential approx. +12 mV), the zeta potential of the system shifted to more negative values which are saturated at ≥2.5 µg/mL SP guided by the orange curve. Compared with SP alone, PF4/SP complexes exhibited a higher negative zetapotential (Δ_ZP_ ≈ −10 mV). (**C**) SP does not cause significant change when interacting with human IgG isolated from healthy donors (a negative control), whereas (**D**) it causes a strong reduction in the negative zeta potential of ACE2 (a positive control). (**E**) Equilibrium dissociation constant (K_D_) of PF4–SP binding determined by isothermal spectral shift analysis. Data are shown as mean and SD (*n* = 2−3).

**Figure 3 biomolecules-14-00245-f003:**
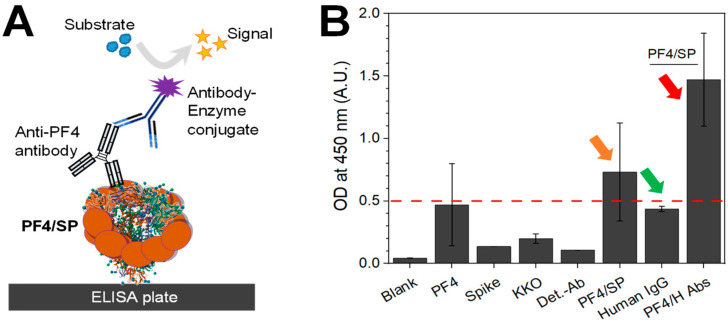
Binding of SARS-CoV-2 SP to PF4 confirmed by ELISA. (**A**) Cartoon shows set up for detection of bound PF4 to SP immobilized on ELISA plate. (**B**) PF4/SP binding (orange arrow) shows higher optical density (OD) values than controls (below OD cut-off 0.5, red line). PF4/SP complexes facilitate the binding of PF4/H antibodies in three well-characterized HIT sera (OD around 1.5, red arrow), but they do not allow for binding of human IgG control in healthy donnors (OD < 0.5, green arrow). Data are shown as mean and SD (*n* = 2–3).

**Figure 4 biomolecules-14-00245-f004:**
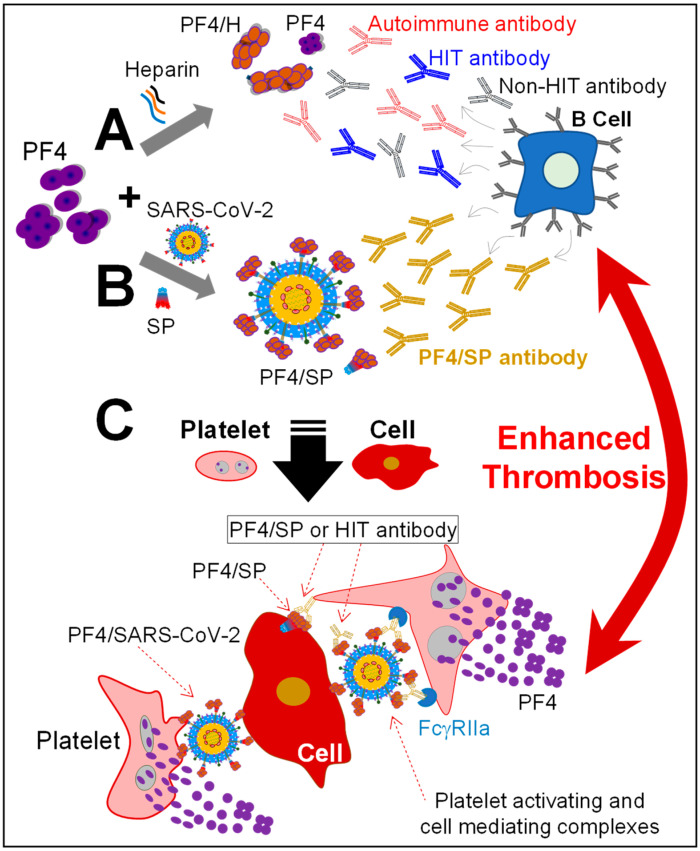
The proposed mechanism for the generation of PF4-antibodies in HIT-COVID-19 system and their role in thrombosis. (**A**) The known mechanism for the formation of anti-PF4/H antibodies (by immune cells, like B-cells) against PF4/H complexes in HIT. (**B**) Proposed mechanism for the development of PF4-antibodies against PF4/virus complexes in SARS-CoV-2 infected patients. (**C**) The synergistic COVID-HIT combination among immune complexes including PF4/virus, PF4/H, and PF4 antibodies; PF4/virus/antibodies cause cross-linking among platelets/cells and activate platelets via FcγRIIa, resulting in severe thrombotic thrombocytopenia. Activated platelets release a large amount of PF4 that again stimulates the formation of PF4-antibodies and, thus, enhances thrombosis formation over time (red arrow).

**Table 1 biomolecules-14-00245-t001:** Samples tested in ELISA.

Adding Material	Blank	Spike Control (SpikeCon)	KKO Control (KKOCon)	Detection Ab Control (Det-AbCon)	PF4/SP
SP	-	+	+	-	+
PF4	-	-	-	-	+
KKO	-	-	+	-	+
Detection Ab	-	+	+	+	+

- or + indicate no binding or binding, respectively.

## Data Availability

The original contributions presented in the study are included in the article, further inquiries can be directed to the corresponding author.

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
