# Peer review of "The Binding of the SARS-CoV-2 Spike Protein to Platelet Factor 4: A Proposed Mechanism for the Generation of Pathogenic Antibodies"

_biomolecules, 2024, doi:10.3390/biom14030245_

Round 1
Reviewer 1 Report
Comments and Suggestions for Authors
In their present study, Nguyen et al. employed multiple methods, including quartz crystal microbalance, dynamic light scattering, isothermal spectral shift analysis, and enzyme-linked immunosorbent assay to identify and verify the interaction between SP and PF4, potentially precipitating immunogenic platelet activation.
1) It is prudent to repeat certain experiments to bolster the veracity of the conclusions. In the context of QCM and DLS analyses, the findings are presented through merely 2-3 repetitions. Moreover, although there are three replicates in the ELISA-confirmed experiment, the substantial intra-group variability (as denoted by the error bars) suggests a lack of stability in the results.
2) Beyond substantiating the binding, an in-depth exploration of which component of SP interacts with PF4 could be invaluable, potentially shedding light on future drug design.
A few minor concerns are also necessitated:
1) In Figure 1B, it is essential to include data on the frequency variations of PF4 and SP independently, to serve as a comparison.
2) Additionally, in Figure 3B, the data of PF4 should be included.
Author Response
Dear Reviewer,
Thank you for dedicating time to review our manuscript and for your valuable comments and suggestions. Enclosed is our response to your insightful feedback.
We trust that our extensive revisions address your concerns and enhance the quality of the manuscript.
Best regards,
Thi Huong Nguyen

Reviewer 2 Report
Comments and Suggestions for Authors
The manuscript reports the binding of SARS-Cov-2 spike protein to Platelet Factor 4 (PF4) under in vitro conditions. The study results obtained using the 4 methods indicate the formation of a complex between the spike protein and PF4, resulting in conformational changes in PF4 that allow the binding of anti-PF4/heparin antibodies. In addition, based on the results, the Authors suggest a mechanism leading to the production of pathological anti-PAF4 antibodies in patients.
The aim of the study is very interesting, especially given the search for mechanisms responsible for thrombotic complications in COVID-19, but the Authors do not emphasize the novelty of the study. The formation of PF4-spike protein complexes has been previously described. Evidence of conformational changes in PF4 in the PF4-SP complex that enable the binding of anti-PF4/H antibodies is insufficient. Statistical analysis of the results is lacking and the conclusions drawn are based on 2-3 replicates. In the ELISA, binding of anti-PF4/H to the PF4-SP complex should be compared to the control with anti-PF4/H (instead of to human IgG).
The Discussion is too long, it could rather be a fragment from a review publication. A discussion of own main results is lost throughout the discussion text.
The presentation of the results should be described in the figure legend (mean and SD).
Author Response

(The authors gave the same response as above.)

Reviewer 3 Report
Comments and Suggestions for Authors
The authors investigated the interaction between the SARS-CoV-2 spike protein (SP) and platelet factor 4 (PF4), proposing a mechanism for the formation of pathogenic PF4 antibodies in COVID-19 and vaccine-induced thrombotic thrombocytopenia (VITT) patients. Using various techniques like quartz crystal microbalance, dynamic light scattering, and enzyme-linked immunosorbent assay, the study demonstrated that SP binds to PF4, altering its structure and leading to the formation of PF4/SP complexes. These complexes can trigger the immune system to produce PF4 antibodies, contributing to severe thrombocytopenia symptoms. The findings suggested potential implications for the diagnosis and treatment of severe COVID-19 and VITT cases, highlighting the need for better detection and management strategies for PF4 antibodies in these conditions.
1. Could the authors conduct a structural analysis using techniques such as X-ray crystallography or nuclear magnetic resonance (NMR) to elucidate the molecular-level interaction between PF4 and the spike protein?
2. Considering the rapid evolution of the virus and the emergence of diverse SARS-CoV-2 variants, it would be beneficial for the authors to clarify which specific spike protein variant was utilized in this study. Future research would greatly benefit from specifying and incorporating a range of spike protein variants to yield more comprehensive and relevant insights. Such an approach would significantly enhance our understanding of the variant-specific interactions with platelet factor 4 (PF4). I recommend that the authors discuss this aspect to strengthen the study's applicability and relevance in the context of the evolving pandemic.
Author Response

(The authors gave the same response as above.)

Round 2
Reviewer 1 Report
Comments and Suggestions for Authors
This revised manuscript can be accepted for publication.
Author Response
Dear Reviewer,
We thank you for reviewing our manuscript and providing helpful comments/suggestions so that the manuscript is significantly improved.
On behalf of all authors,
Dr. Thi Huong Nguyen
Reviewer 2 Report
Comments and Suggestions for Authors
I have only one comment:
In the legends to figures instead "Mean and SD were obtained from 2-3 repetitions." should be rather "Data shown as mean (n=2) or mean and SD (n=3)"
Author Response
Dear Reviewer,
We thank you very much for reviewing our manuscript and providing us with helpful comments/suggestions so that the manuscript is now significantly improved.
In response to your comment, we have now modified the Figure legends (Fig. 1-3) in the way you suggested. Figure legends now state:
Data are shown as mean and SD (n= 2-3).
Best wishes,
Dr. Thi Huong Nguyen